# Atomic intercalation to measure adhesion of graphene on graphite

Jun Wang[1], Dan C. Sorescu[2], Seokmin Jeon[1], Alexei Belianinov[1], Sergei V. Kalinin[1], Arthur P. Baddorf[1] & Petro Maksymovych[1]

The interest in mechanical properties of two-dimensional materials has emerged in light of new device concepts taking advantage of flexing, adhesion and friction. Here we demonstrate an effective method to measure adhesion of graphene atop highly ordered pyrolytic graphite, utilizing atomic-scale 'blisters' created in the top layer by neon atom intercalates. Detailed analysis of scanning tunnelling microscopy images is used to reconstruct atomic positions and the strain map within the deformed graphene layer, and demonstrate the tip-induced subsurface translation of neon atoms. We invoke an analytical model, originally devised for graphene macroscopic deformations, to determine the graphite adhesion energy of $0.221 \pm 0.011\,\mathrm{J\,m^{-2}}$. This value is in excellent agreement with reported macroscopic values and our atomistic simulations. This implies mechanical properties of graphene scale down to a few-nanometre length. The simplicity of our method provides a unique opportunity to investigate the local variability of nanomechanical properties in layered materials.

[1] Center for Nanophase Materials Sciences, Oak Ridge National Laboratory, Oak Ridge, Tennessee 37831, USA. [2] Research & Innovation Center, National Energy Technology Laboratory, U.S. Department of Energy, Pittsburgh, Pennsylvania 15236, USA. Correspondence and requests for materials should be addressed to P.M. (email: maksymovychp@ornl.gov).

Adhesion is a key parameter in fabrication of next generation nanoscale mechanical resonators based on two-dimensional (2D) materials[1,2], and a rapidly growing family of 2D heterostructures[3–8]. Various methods to measure the adhesion energy have been tested and reported previously[9–12]. However, nanoscale measurements of these properties are generally very challenging due to the weak interlayer bonding in layered materials[13]. Even for graphite, a direct measurement of interlayer adhesion energy is still limited[13,14], despite many theoretical predictions[15–17]. Only recently, an adhesion energy of $0.227\,\mathrm{J\,m^{-2}}$ was determined from direct measurements of mesoscopic graphene contacts, using the shearing of the individual graphitic mesa structures[13]. Several studies reported the utilization of graphene micron-sized blisters or bubbles created by intercalated nanoparticles at the hetero-interface between graphene and the $SiO_2$ substrate[18], or by inflation of pre-made microcavities on the $SiO_2$ substrate to form graphene bubbles[9]. Nevertheless, all such blisters are of relatively large scale, and may not be readily compatible with van der Waals heterostructures.

Here we demonstrate that the 'blister' method can be scaled down to 1–2 nm by using atomic-scale intercalation. We intercalate neon atoms into graphite, which in turn deform enclosing graphene sheets to create atomic-scale blisters on the surface. Detailed experimental analysis of these atomic blisters leads to a direct estimate of adhesion energy of $\sim 0.221\,\mathrm{J\,m^{-2}}$ between the first-layer graphene and graphite bulk, which is closely comparable with the recent direct measurement of adhesion energy between mesoscopic graphite contacts[13]. At the same time we measure the strain within the blister area by direct topographic analysis, and we prove the feasibility to displace buried intercalates. Our experimental results for adhesion energy, the local topographic characteristics of the blisters as well as the low diffusion barrier of intercalates are strongly supported by density functional theory (DFT) calculations.

## Results

**STM characterization for atomic blisters.** Representative scanning tunnelling microscopy (STM) images of graphene blisters on highly ordered pyrolytic graphite (HOPG) are shown in Fig. 1a,b. The blister in Fig. 1b has a nearly Gaussian shape with the height $h$ of $\sim 0.14$ nm and full-width at half-maximum of $\sim 0.93$ nm, respectively (Fig. 1d). For reference, the atomic radius of a neon atom is $\sim 38$ pm, while that of carbon is $\sim 67$ pm (ref. 19). The van der Waals radius is similar for both atoms (neon $\sim 154$ pm and carbon $\sim 170$ pm)[20]. Taking into consideration the approximate lateral size of the blister of $\sim 1.9$ nm (at the base in Fig. 1d), we estimate that only a couple of neon atoms are intercalated in the blister, while the strain in the top graphene sheet is delocalized over $\sim 60$ atoms around the neon intercalates. Besides the atomic blisters, other defects are observed such as holes created by ion bombardment, as shown in the STM image of Fig. 1a (enclosed by squares). These point defects are due to missing carbon atoms. For the purpose of this article, we only focus on the blisters of the chemically undisrupted top graphene sheet. Since we are working with a grounded sample, and both graphite and graphene are electronically conducting materials, Ne impurity is charge neutral. Neutralization of the low energy $Ne^+$ is well-known for metal surfaces with work functions in the range from 3–5.5 eV (refs 21,22). Our DFT calculations further confirm that there is a minimal ($<0.05\,e$) charge transfer between the intercalated Ne atoms and surrounding carbon atoms of the graphite, further asserting that we can treat Ne intercalates as essentially neutral atoms.

**Lattice strain caused by the intercalate.** To map out atomic positions and their local neighbourhoods, we employed image processing as illustrated in Fig. 1e,f. Figure 1e indicates the $x$–$y$–$z$ coordinate map of atom centres extracted from Fig. 1b. Each dot represents the carbon atoms observed by STM (every other carbon atom on HOPG), and the blister region can be clearly identified by the height difference (in Fig. 1e). Furthermore, we estimated the tensile strain in the blister by mapping out the expansion of the C..C distance, as seen in Fig. 1f. The colour of each point here represents the percentage of expansion of C..C distance (between every other carbon atoms on HOPG) in the blister region compared with the one in the undistorted graphene region. The C..C distances are expanded from $\sim 1\%$ up to a remarkable $\sim 25\%$ compared with the undistorted graphene, with an average increase of $\sim 6\%$. Several methods can be used to estimate the strain. For example, by using the model for the mechanics of graphene bubbles developed by Yue et al.[10] (equations (18) and (19) in ref. 10), we obtain the maximum strain for both radial and circumferential strains of $\sim 3\%$ at the central deflection of the blister as illustrated in Fig. 1b. Alternatively, based on the definition of Georgiou et al.[23] in which the strain is determined as the percentage increase between the length of the arc and the width of the bubble, and in combination with the line-profile indicated in Fig. 1d and the strain map from Fig. 1f, we estimate a strain of $\sim 1$–3% for the blister in Fig. 1b. Overall, this analysis confirms that in the blister structure, the top layer of graphene is stretched to accommodate subsurface intercalated atoms.

**Analytical models.** Several models have been used before by Yue et al.[10] to analyse the adhesion energy between graphite layers: the membrane model, the nonlinear plate model and the linear plate model. The membrane model is applicable for large graphene bubbles ($h > 10$ nm). The nonlinear plate model is generally more accurate and suitable for all sized bubbles. Within this model, the adhesion energy ($\Gamma$) of the graphene blister having a characteristic radius $a$ and a height $h$, (Fig. 1d) is obtained from the equation[10]:

$$\Gamma = \frac{80\mu E_{2D} h^4}{3a^4} + \frac{32D h^2}{a^4} \tag{1}$$

Here $E_{2D}$ is the 2D Young's modulus of graphene; $D$ is the bending stiffness, which for macroscopic graphene is $D = 0.238$ nN nm (or equivalently 1.5 eV)[10,24]; while $\mu$ represents a function of the Poisson ratio $v$:

$$\mu = \left(7,505 + 4,250v - 2,791v^2\right) / \left(211,680\left(1 - v^2\right)\right) \tag{2}$$

The linear plate model is an approximation of the nonlinear plate model for small sized bubbles generally having $h < 0.3$ nm and $h \ll a$. In such cases the first term in equation (1) can be ignored and the adhesion energy is expressed as

$$\Gamma = \frac{32D h^2}{a^4} \tag{3}$$

which is simply a function of the blister's height $h$ and radius $a$. Although this linear plate model was derived for a small blister, it's validity for atomic scale is not clear a priori. In this study, we test this model extension by using the simplified version of the linear plate model given in equation (3) to determine the adhesion energy of graphene to graphite. For this purpose, to quantitatively identify the height $h$ and the radius $a$ of the atomic blister, we fit the STM line-profile with a Gaussian function:

$$z(r) = z_0 + h \times \exp\left[-\left(\frac{r - r_0}{\sqrt{2}\sigma}\right)^2\right] \tag{4}$$

as indicated by the red curve in Fig. 1d.

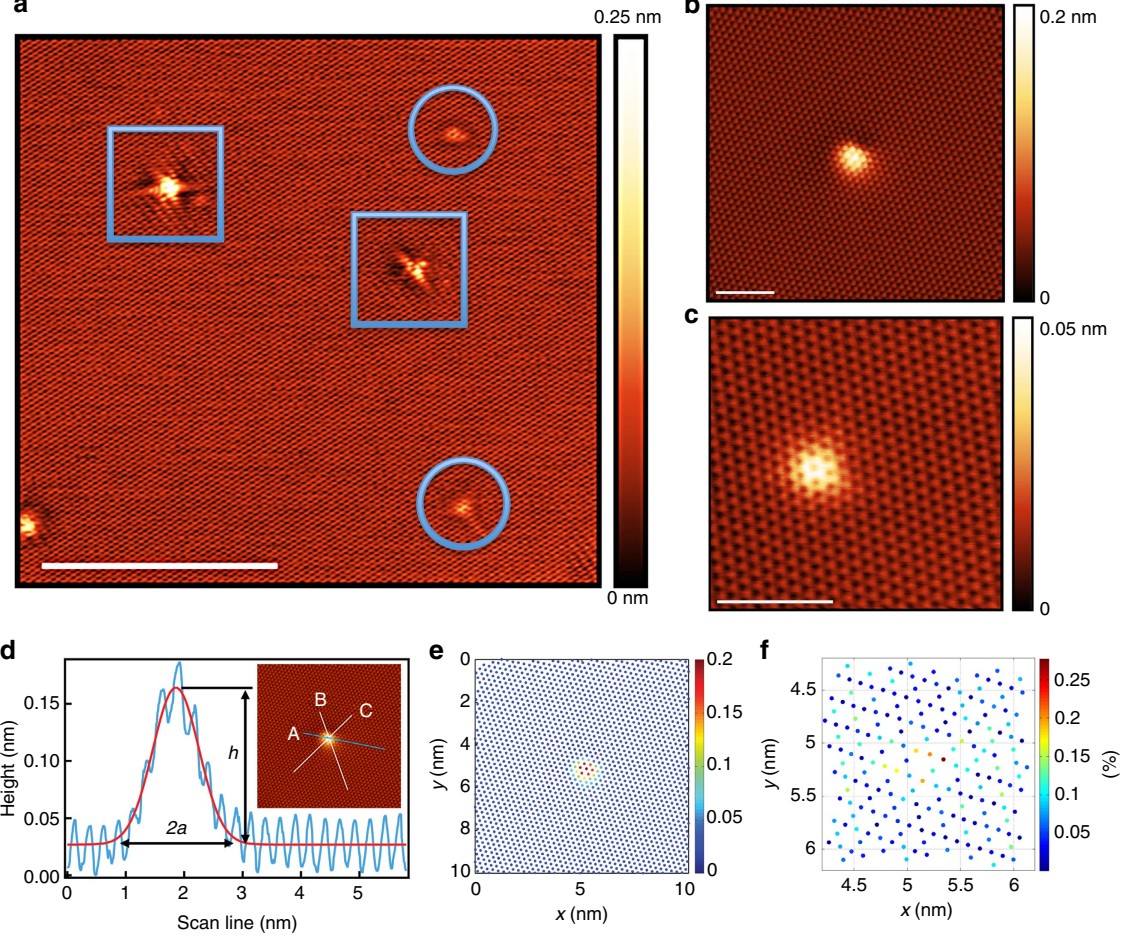

**Figure 1 | Atomic blisters created by ion sputtering on graphite. (a)** A large scale atomically resolved STM image, showing several defects and blisters created by low energy neon ion sputtering. The blisters are highlighted by circles, while the point defects are highlighted by squares ($U_{\text{sample}} = 0.3$ V, $I_t = 0.2$ nA, $T = 77$ K). Scale bar, 10 nm. **(b)** Atomically resolved STM image of a nearly circular blister on graphite (sample bias $U_{\text{sample}} = 0.3$ V, tunnelling current $I_t = 0.2$ nA, temperature $T = 77$ K). Scale bar, 2 nm. **(c)** Another type of blister with a somewhat triangular outline as resolved by STM ($U_{\text{sample}} = 0.3$ V, $I_t = 0.2$ nA, $T = 77$ K). Scale bar, 2 nm. **(d)** A representative line-profile of the atomic blister in Fig. 1b (taken along the scan line A, inset) indicated by the blue curve; the red curve is a Gaussian fit; $h$ indicates the height of the blister, while $2a$ represents the approximate width of the blister at the base; scan lines B and C in the inset figure are additional line profiles as explained in the text. **(e)** The $x$–$y$–$z$ coordinate map of the atom centres extracted from Fig. 1b. The $z$ scale bar is in units of nm. **(f)** The atomic-scale strain map between the carbon atoms in the blister region: The colour of each dot represents the percentage of length increase between every other carbon atom of graphene (hereafter denoted as C..C) in the blister region compared with the one in the undistorted region, that is, [d(C..C)-$d_0$(C..C)]/$d_0$(C..C), where $d_0$(C..C) is the C..C distance of the undistorted region.

**Measurements of adhesion energy.** We measured a number of line profiles over the blister by repeated rotation of the one-dimensional cross-section (such as the lines A–C in Fig. 1d inset), fitted them by a Gaussian function (4) to find the radius $a$ and the height $h$, and obtained the adhesion energy via equation (3). Figure 2 (data sets in red) shows the calculated adhesion energies with respect to different measurements. The mean of the adhesion energy obtained is $0.221 \pm 0.011$ J m$^{-2}$ (Fig. 2). Our result is very close to the recently reported graphite adhesion energy of 0.227 J m$^{-2}$ (ref. 13). Regarding the errors, the adhesion energy is proportional to $h^2/a^4$, while a Gaussian shape did not always represent the blister profile. Therefore, even picometre-scale variation of $a$ and $h$ cause a relatively wide scatter of the adhesion energy. Nevertheless, our method provides a simple route to measure the adhesion energy and our results are closely matching other reported values[13,14,25,26]. Our study also demonstrates that the linear plate model developed by Yue et al.[10] can provide estimation for the adhesion energy even for atomic scaled blisters. At these scales, the models can be compared directly to results from first principles or molecular dynamics simulations[15–17,27–29].

**Pressure inside the blister.** Another physical property of interest is the pressure on the top-layer graphene exerted by the buried Ne atoms. This quantity was calculated using the corresponding expressions from the nonlinear plate model[10],

$$p = \frac{64\mu h}{a^4}\left(E_{2D}h^2 + D\right) \tag{5}$$

where $p$ is the pressure in Pascal (Pa) unit, $E_{2D}$ is the 2D Young's modulus of graphene, $D$ is the bending stiffness, $a$ and $h$ are radius and height of the blister; while $\mu$ represents the function of the Poisson ratio $v$ given in equation (2). For $E_{2D} = 0.323$ TPa nm and $v = 0.179$, we estimate a pressure of 5.4 GPa within the nonlinear plate model (an estimate for the membrane model is 6.3 GPa). For reference, the pressure of the macroscopic blisters determined in the work of Yue et al.[10] was a thousand-fold smaller ranging between 1 and 2 MPa. A high value of the pressure in our experiments reflects the incompressibility of the individual Ne atoms.

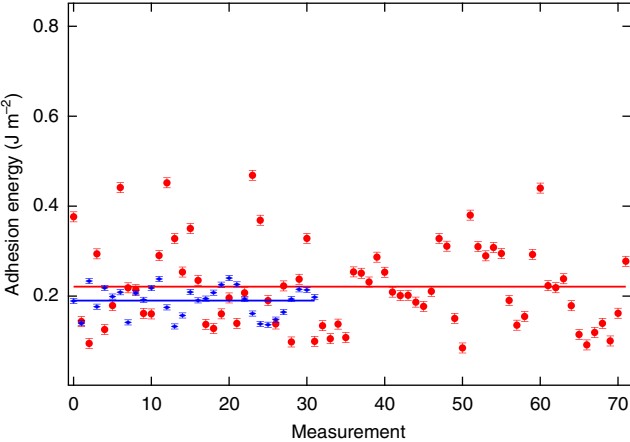

**Figure 2 | Nanoscale measurement of the adhesion energy of graphite.** Data sets in red: the measured adhesion energy ($\Gamma$) for different cross-sections of the blister in Fig. 1b. The mean value of the adhesion energy between graphene and graphite interlayers is $0.221 \pm 0.011 \, \mathrm{J\,m^{-2}}$ (indicated by the red line). Data sets in blue: the measured adhesion energy ($\Gamma$) for different cross-sections of the blister in Fig. 1c, after properly scaling the corrugation as explained in the text. The mean value of the adhesion energy in this case is $\sim 0.19 \, \mathrm{J\,m^{-2}}$ (indicated by the blue line). All error bars are obtained as standard errors, that is, the standard deviation divided by the square root of the number of measurement.

**Density functional theory verification**. Our approach is further validated by atomistic modelling using DFT calculations. Figure 3a presents the results of simulations and illustrates top and side views of a system of four neon atoms intercalated into a graphene bilayer. A corresponding simulated STM image (Fig. 3b) bears close resemblance to experiment. We have simulated 1–4 intercalated Ne atoms and then used a 2D-Gaussian function to fit the distribution of atomic coordinates. The variation in the height of the blister with the number of neon atoms incorporated in the graphene bilayer is plotted in Fig. 3c. The results are within the range of STM measured values (see Fig. 1d). Figure 3d shows the dependence of the adhesion energy on the number of neon atoms. The squares are from DFT calculations, while circles represent the results from the linear plate model applied to the calculated structures. These values are well within the range of the ones derived from experiments and can be extrapolated to zero intercalates leading to a value for adhesion energy of $\sim 0.235 \, \mathrm{J\,m^{-2}}$. Notably the adhesion energy does decrease with the increase in the number of intercalated Ne atoms. These results suggest that the local adhesion of graphene layers can be substantially influenced by an increase in the number of intercalated neon atoms, particularly if they are localized in the same region. It is also expected that further increase in concentration of intercalated neon atoms can induce local exfoliation of the top graphene layer.

**Manipulation of blisters**. Our experiments further indicate the feasibility of tip-induced manipulation of subsurface Ne atoms. Figure 4a,b demonstrate the effect of several STM scans at decreased tip–surface distance (tip–surface distance was reduced by decreasing the bias down to 10 mV, while increasing the tunnelling current up to 3 nA). The blister highlighted by a circle in Fig. 4a has been moved out of the current image frame as seen in Fig. 4b. We calculated the minimum energy diffusion pathway of a single Ne atom in the gap of the graphene bilayer

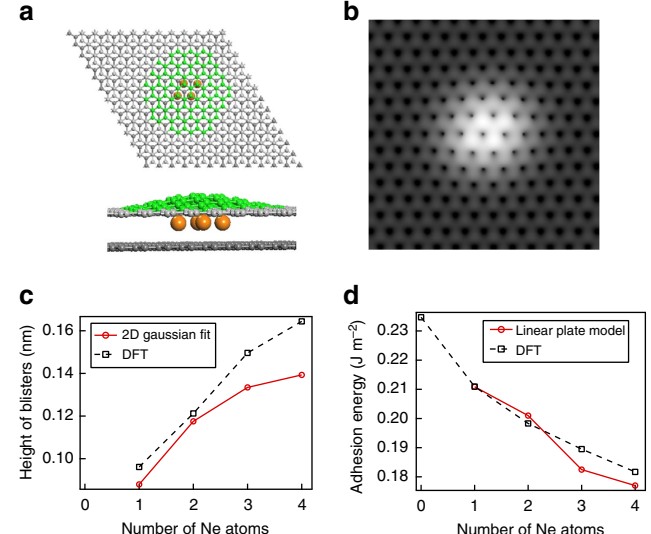

**Figure 3 | DFT simulation of atomic blisters and adhesion energy of graphite.** (**a**) DFT model of four Ne atoms intercalated between two graphene layers (top and side views); green atoms indicate the deformed graphene regions. (**b**) A simulated STM image at a bias of $V = 0.3 \, \mathrm{V}$ for the graphene blister with four Ne atoms intercalated between a bilayer graphene. (**c**) Blister height as a function of the number of Ne atoms as determined from DFT calculations and from the 2D-Gausssian profile fit to the computed topography. (**d**) Adhesion energy of the top graphene layer as a function of the number of Ne atoms as derived from both DFT calculations and the linear plate model applied to the computed topography. The adhesion energy in DFT calculations was determined as the adsorption energy (per surface area) of the top graphene layer on the bottom graphene layer having from 1–4 adsorbed Ne atoms in the gap between layers.

between neighbour energy minima using the nudged elastic band method. As seen from Fig. 4c, a relative modest barrier of only 0.14 eV is encountered by the diffusing Ne atom in the gap that strongly supports the feasibility of manipulating subsurface intercalates.

**Smaller height blister versus deeper layer intercalation**. One may question the applicability of STM to measure topographic height. In fact, it is well-known that the atomic corrugation of graphite in STM images depends on the tip conditions[30–37]. In earlier studies[32–34], even giant atomic corrugations of graphite (1–7 Å, or even larger up to 24 Å) have been reported in STM images, which were often related to tip-induced elastic deformations or to other experimental limitations such as contaminations at ambient conditions[32–34]. Tersoff *et al.*[37] demonstrated theoretically that the atomic corrugation of graphite in STM images is close to 0.3 Å or less, which is similar to the corrugation ($\sim 0.35$ Å) obtained in our case as illustrated in Fig. 1b. In combination with our theoretical calculations of the blister height in Fig. 3c, we believe that $\sim 0.3$ Å represents an accurate value of the graphite atomic corrugation and therefore the profile measurement in Fig. 1b represents accurately the blister's height.

In a few cases (see Fig. 1c and the corresponding line profile in Supplementary Fig. 2) the measured graphite atomic corrugation was smaller, $\sim 0.1$ Å. In such a case the blister, likewise, has a smaller height of $\sim 0.03$ nm. A proper scale (with a factor of 3.5) was applied to the z-height in this case to make the atomic corrugation in Fig. 1c close to the one in Fig. 1b. Upon applying

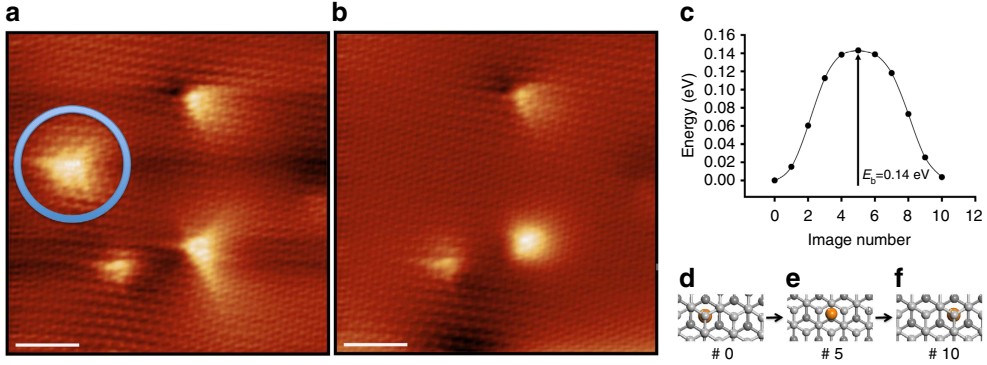

**Figure 4 | Manipulation and diffusion of atomic blisters and the calculated diffusion energy barrier.** (**a**) Atomically resolved STM image captures the blister (enclosed by a circle) and other point defects ($U_{sample} = 0.2\,V$, $I_t = 1\,nA$, $T = 4.3\,K$). Scale bar, 2 nm. (**b**) The STM image of the same region as of **a** shows the blister disappeared (that is, moved out of the image frame) after several near contact-region scans, leaving the flat and intact graphene sheet ($U_{sample} = 0.2\,V$, $I_t = 1\,nA$, $T = 4.3\,K$). Scale bar, 2 nm. (**c**) Minimum energy pathway for Ne atom diffusion in the gap of graphene bilayer between two successive local minima. The initial, transition and final state configurations are indicated in **d**–**f**.

the linear plate model to the rescaled blister profile, a mean adhesion energy of $\sim 0.19\,J\,m^{-2}$ was obtained out of 32 total measurements of the rescaled blister (data sets in blue in Fig. 2).

On the other hand, the smaller height of the blister may be due to Ne atoms intercalated into the deeper layers. Although we believe that this is not the case for the blister in Fig. 1c (see Supplementary Note 1), deeper layer intercalation is possible judging by the stability of the final state. We carried out DFT calculations of deep intercalates and determined a substantial reduction of the blister height when intercalating beyond the first layer (see Supplementary Note 2 and Supplementary Figs 3, 4 and 6). Experimentally we do detect occasional small protrusions after $Ne^+$ sputtering, which may correspond to such deep intercalates (see Supplementary Fig. 1 and Supplementary Note 2). However, direct estimates of the adhesion energy as described above should be applied only to the case when the Ne atoms are located immediately underneath the surface (see Supplementary Fig. 5).

**Nc-AFM characterization for atomic blisters.** An alternative, more direct method for the measurement of the blister topography can be obtained using non-contact atomic force microscopy (nc-AFM). Nc-AFM detects short-range forces, and the nc-AFM image provides direction information about the corresponding tip–surface interaction[38]. The atomically resolved nc-AFM image (Fig. 5b) shows protruding blisters that compare favourably to those observed in STM over the same blister. Line profiles comparison (Fig. 5c) does indicate small height variations between STM (Fig. 5a) and nc-AFM (Fig. 5b) measurements. (STM measured height of $\sim 0.16$ nm versus nc-AFM measured height of $\sim 0.10$ nm). However, the overall shape of the blister is consistent in both measurements, convincingly indicating that the topographic lattice deformations are induced by the Ne atom intercalate.

In conclusion, we demonstrate a direct method to measure the adhesion energy of graphene on graphite, by intercalating inert gas atoms between graphene sheets. Measurements are based on the shape and heights of the graphene blisters at the atomic scale and analysed using analytical methods (linear plate models) and DFT calculations. Our results are in good agreement with recently reported direct measurements of adhesion energy for graphite. Local topographic analysis also provides detailed atomic information related to the graphene blisters, such as local strain and the mobility of subsurface noble gas atoms. We envision that this methodology can be applied to other layered materials to

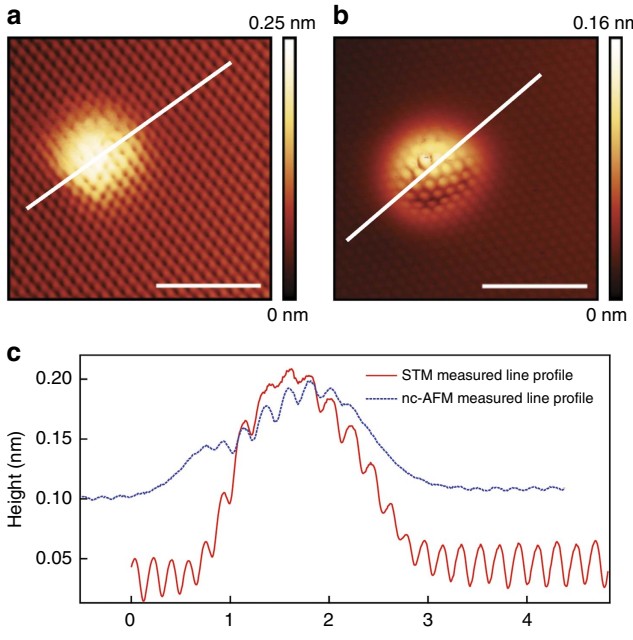

**Figure 5 | Comparison between STM and nc-AFM measurements of the same graphene blister.** (**a**) The STM image was acquired at a bias of $U_{sample} = 0.63\,V$, a tunnelling current of $I_t = 0.1\,nA$ and $T = 4.3\,K$. Scale bar, 2 nm. (**b**) The nc-AFM image was obtained at a frequency shift of $f = -0.875\,Hz$, using an oscillation amplitude $A = 165\,pm$ at the resonance frequency of $\sim 992\,kHz$ and $T = 4.3\,K$. Scale bar, 2 nm. (**c**) The corresponding line profiles for the scan lines in **a** and **b**, the red solid curve is the STM measurement while the blue dashed curve is the nc-AFM measurement.

estimate the adhesion energy and the related elastic mechanical properties at atomic level. Moreover, intercalation of noble gas atoms may provide a feasible pathway to create electronically interesting 'Gaussian' impurities[39,40].

## Methods

**Experiments.** An HOPG sample (Grade SPI-1) was purchased from SPI Supplies. The experiments were conducted by using the SPECS Joule-Thomson (JT) cryogenic STM/AFM in an ultrahigh vacuum chamber with a base pressure $\leq 1 \times 10^{-10}$ mbar. The nc-AFM measurements were carried out in the same system with a SPECS Kolibri sensor. HOPG was freshly cleaved by the scotch

tape method and quickly transferred into ultrahigh vacuum. The HOPG was subsequently annealed up to 500 °C for ~20 min. The blisters and associated defects on HOPG were created by briefly sputtering the sample with $Ne^+$ ions (0.11 kV, $5.5 \times 10^{-8}$ mbar, for ~20 s). These sputtering conditions are necessary for the reproducibility of atomic sized blisters. Lighter neon was chosen over argon to minimize creation of point defects by sputtering. STM images were taken after cooling the sample to both liquid nitrogen (77 K) and liquid helium (4.3 K) temperatures. The STM profile measurements over different blisters taken at 77 and 4.3 K yield consistent results, suggesting in this temperature range adhesion energy shows little change, as may be expected (see Supplementary Note 3). Atomic-scale analysis was performed using a methodology described earlier[41,42]. Atomic $(x, y)$ positions were determined with subpixel precision using a combination of image processing and fitting routines. After the atomic positions were established, a survey of the six nearest neighbours was conducted to obtain the data in Fig. 1f.

**Theory.** The adsorption properties of Ne atoms in the gap between two parallel graphene layers were investigated based on DFT calculations using Vienna *ab initio* simulation package (VASP)[43,44] in conjunction with periodic slab models. The computations used the Perdew-Burke-Ernzerhof (PBE)[45] exchange correlation functional and the projector-augmented wave method of Blöchl[46,47]. The standard PBE functional was corrected to include long-range dispersion interactions using the Tkatchenko and Scheffler (TS) method[48]. A cutoff energy of 500 eV was used for the plane-wave basis set. The slab model used in calculations consists of a $12 \times 12$ graphene bilayer separated in the vertical direction by a vacuum width of 16 Å. After the initial optimization of the bare graphene bilayer the bottom layer was kept fixed in subsequent Ne adsorption calculations while the top layer was fully relaxed. Additional information about the energetics of single Ne atom diffusion in the gap of graphene has been calculated using the climbing image-nudged elastic band method[49,50]. For selective adsorption configurations the corresponding STM images were calculated using the Tersoff–Hamman approach with an energetic interval around the Fermi level similar to the one used experimentally[51]. The sampling of Brillouin zone was performed using a single Γ-point in the case structural optimizations while a Monkhorst-Pack scheme[52] with a grid mesh of 0.005 Å$^{-1}$ k-point separation was used for STM image calculations.

**Data availability.** The data that support the findings of this study are available from the corresponding author upon request.

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

## Acknowledgements

Experiments were conducted at the Center for Nanophase Materials Sciences (CNMS), which is a DOE Office of Science User Facility. Image analysis was carried out in collaboration with the Institute for Functional Imaging of Materials at the Oak Ridge National Laboratory. Computations were conducted at National Energy Technology Laboratory on Joule supercomputer system.

## Author contributions

J.W., A.P.B and P.M. designed the research. J.W., S.J. and P.M. conducted the experiments. A.B. and S.V.K. provided guidance in the data analysis. J.W. and P.M. performed the data analysis. D.C.S. conducted the DFT calculations. J.W. and P.M. wrote the paper with inputs and corrections from all authors.

## Additional information

**Competing financial interests:** The authors declare no competing financial interests.

