## [Peer Review File · Nature Communications]

Reviewers' comments:

Reviewer #1 (Remarks to the Author):

The authors report on an intriguing experimental study of a local "exfoliation" of a graphene sheet induced by the intercalation of Ne atoms. Combining topographic structure data obtained from high resolution STM images with mechanical deformation models and DFT calculations the interface energy in nm scale structures is accurately measured. This is a remarkable achievement of this very elegant experiment. In addition, it is shown that the Ne atoms stabilize a huge disjoining pressure of 5 GPa and that the Ne defects can be easily translated by the effect of the scanning tip without destroying the graphene lattice.

Overall the manuscript is well organized and clearly written and the relevant literature is cited. However, there are a few points that should be clarified:

(1) It is not clear to me at what depth the Ne atoms are intercalated. For me it reads that it is implicitly assumed that the atoms sit at the interface below the first graphene sheet. This is not obvious to me and the authors should clarify this point.

(2) Related to (1) the authors write that the triangular shaped blisters are in general 3 times less tall than the circular ones. Couldn't this be due to the fact that the Ne atoms are intercalated deeper in the HOPG substrate.

(3) The paragraph discussing the Van der Waals interaction on page 12 is confusing to me. What is it precisely the authors intend to say. Likewise the message in the statement at the end of the paragraph is entirely obscure to me.

(4) The term Ne atom diffusion is misleading. The authors clearly say that they had to use imaging conditions corresponding to a short tunneling distance for observing this effect. I would rather use the term tip-induced translation in this context.

Reviewer #2 (Remarks to the Author):

The authors measured the adhesion energy of graphene atop highly ordered pyrolytic graphite (HOPG). They used STM images to reconstruct the atomic-level displacement and strain map of the graphene blisters created by ion-implanted intercalates. The adhesion energy of 0.221 ± 0.011 J/m² was extracted based on an adapted plate model for the blister. This value is in close agreement with a recent report as well as the first principles calculation result.

Overall, the authors have performed careful experiments and analysis. The results are interesting and useful. The paper is clearly written. However, my major concern is on the novelty and significance of the outcome of this work. It seems to be the measured adhesion energy of graphene that has been recently reported in a Science paper for mesoscopic graphite contacts (ref. [13]). Since the adhesion energy is a fundamental material property, it is expected to remain unchanged at the nanoscale, as verified by this work.

In addition, it is not clear whether the implanted neon atoms will penetrate into several graphite layers. Is it possible that the multi-layered intercalates collectively produce the surface clusters?

In the abstract, what is the meaning for "measurement of elastic adhesion"? The "elastic adhesion" is not a commonly used term. In other words, what is "inelastic adhesion"?

Reviewer #3 (Remarks to the Author):

As the author pointed out, the nano-scale measurement of adhesion is of great importance in understanding two dimensional materials. Previously blister tests were adopted to measure the

graphene adhesion; however the measurement was confined to the μm -scale due to the spatial resolution of the atomic-force-microscopy. In this manuscript, the STM based measurement successfully observed the nano-size blister by the Ne atom intercalations. The proposed method was used to correlate the atomistic simulation, and the measured data well agree with the reported values. Furthermore, it is quite interesting that the continuum mechanics model is still valid in atomic-size levels. However, some important details are lacking, and the following questions are raised.

#1. What is the reason for choosing Ne? Would you be able to get the same adhesion energy with Ar intercalation?

#2. As mentioned in the manuscript, Ne+ "ions" were sputtered to create the blisters and associated defects on HOPG. The ions are not inert and may strongly interact with electrons in graphene, greatly affecting the adhesion energy. However, in this manuscript, "inert Ne atom" was assumed not only for the analysis of experimental data, but also for the atomistic simulations. Please comment on this.

#3. It was not clearly explained how the Ne+ ions were intercalated beneath the top graphene layer. Some questions related to this are as follows.

- Isn't it also possible that the Ne+ ions penetrate deeper and end up being intercalated beneath 2nd, 3rd or other layers?

- Where did carbon atoms from the point defects go after being bombarded by Ne+ ions? Any possibility that the intercalated atoms are actually the C atoms instead of Ne atoms? The calculated energy barrier for the Ne atom diffusion is 0.14 eV; how about the energy barrier for the C atom diffusion in the gap of the graphene bilayer? Which atom is more likely to be intercalated in the gap?

#4. The adhesion energy was calculated from the STM image at temperature $T = 77\text{ K}$, which is very low temperature considering real life applications of graphene. The manuscript says that the graphite adhesion energy of $0.221 \pm 0.011\text{ J/m}^2$ determined in this study is in excellent agreement with reported macroscopic values and the atomistic simulations of this study. This reviewer wonders that whether the values being compared were also measured or calculated at the same temperature $T = 77\text{ K}$. Please add some comment on the temperature dependence of the adhesion energy of graphene. Is the adhesion energy at room temperature higher or lower than that at 77 K ?

#5. Can the intercalation and STM measurement be applied on other substrates? For example, can the adhesion energy of transferred graphene on SiO_2 be measured by the method suggested in this study?

#6. Term and symbol should be appropriately used. In case of the E2D, it is the "two dimensional" Young's modulus (the unit is different with the Young's modulus). In case of the adhesion energy, γ (Γ) is suitable rather than E.

#7. How was the blister radius (a) defined in case of the triangular blister? (Fig. 3a)

Part I – Reviewer 1

Overall the manuscript is well organized and clearly written and the relevant literature is cited. However, there are a few points that should be clarified:

Question 1: It is not clear to me at what depth the Ne atoms are intercalated. For me it reads that it is implicitly assumed that the atoms sit at the interface below the first graphene sheet. This is not obvious to me and the authors should clarify this point.

This is indeed a very important point. Because all three reviewers asked a similar question we provide here a common answer to all three corresponding questions. Specifically, we carried out extensive DFT calculations of deep layer intercalation and additional experiments using, non-contact AFM technique for direct measurement of surface topography. Our overall conclusion is that deeper intercalation is in principle possible. It is, however, quite easy to identify the sub-surface intercalates based on the height and curvature of the blisters they create. Most importantly, the direct estimation of the adhesion energy based on the topographic measurement of the surface should be applied only to sub-surface intercalates (as we have done in the original paper). Altogether this additional effort has significantly improved our analysis, and we are thankful to reviewers for their constructive criticism.

We have carried out DFT calculations for Ne intercalated in deeper layers (i.e., between 1st and 2nd, 2nd and 3rd, 3rd and 4th layers) for a 6 layer slab of graphite and the corresponding results are indicated in the revised manuscript in the Supplementary Information. Three conclusions emerged:

- (1) The height of the blister measured at the surface is progressively reduced with increasing the depth of intercalation. The height of 100 pm and above, which corresponds to blisters used in our analysis, is consistent with 1-3 Ne atoms located in the sub-surface layer (Fig. S4). A 3-atom cluster between the 2nd and 3rd layers also yields a blister that is ~ 85 pm in height. All the other configurations, including intercalation beyond the 3rd layer produce much smaller blister heights.
- (2) The curvature of the blister is likewise reduced with increasing intercalation depth. A good way to see this is to calculate the adhesion energy with the linear plate model. As shown in Fig. S5, only the blisters from the sub-surface intercalation yield adhesion energy of 0.15-0.2 J/m², which is consistent with DFT calculations (and close to the macroscopic value). All the other deeper intercalations produce values at least 4 times smaller, due to decreasing curvature of the blister as measured at the top surface.
- (3) We also find that any Ne intercalate distorts both the top and the bottom graphene layers encapsulating the intercalated atom. However, the distortion of the layers in the direction of the open graphite surface is 2-3 times larger (judged by the height, Fig. S6 and Fig. S3) than that in the direction of the graphite bulk. Such deformations on both sides of the Ne intercalate are not accounted for by the plate model, and are a source of systematic underestimation of the adhesion energy. However, as seen in Fig. S5,

the error is quite small relative to true DFT values, and will likely be masked by other experimental errors. At the same time, the trend of reduction of adhesion energy with increasing number of intercalated atoms is reproduced by the plate model (Fig. S5).

- (4) Finally, we included additional experimental data on protrusions with a height of 20-30 pm in Fig. S1. In principle, they could indicate deeper intercalates, but given the uncertainty in their exact structure, such protrusions are not used for direct analysis of adhesion energy.

The following changes were made to the manuscript:

- (a) A new paragraph discussing the properties of deeper intercalates has been added on page 12.*
- (b) A new Fig. 5 comparing non-contact atomic force microscopy and scanning tunneling microscopy has been added in the main text.*
- (c) Detailed computational analysis of deeper intercalates (Fig. S3 through Fig. S6), and a Supplementary Note 2 have been added in Supplementary Information section.*
- (d) A new STM image of possibly deeper intercalates has been included as Fig. S1 in Supplementary Information section.*

- 2. Related to (1) the authors write that the triangular shaped blisters are in general 3 times less tall than the circular ones. Couldn't this be due to the fact that the Ne atoms are intercalated deeper in the HOPG substrate.**

In light of the new calculations discussed above it can be concluded that intercalation of Ne atoms at deeper levels is also possible. However, the peculiarity of this blister with triangular outline is that the constant-current topographic corrugation of the surrounding, unperturbed graphite surface is smaller than in most other STM images. When we rescale the height of the blister in Fig. 1c by the ratio of the z-corrugation between Fig. 1b and Fig. 1c, the blister in Fig. 1c becomes comparable in height to Fig. 1b, and its analysis also produces similar adhesion energy. Our main point of including this data is to show that one should be aware of “artifacts” of the STM topography, mostly due to unknown electronic state of the tip. By normalizing to a certain chosen corrugation may be a way to account for these artifacts. The preferred method would be atomically-resolved non-contact AFM, as we show in the new Fig. 5, which provides a more direct topographic measurement.

On Pages 12-14 of the paper, we added two paragraphs addressing this question. We also added a direct comparison of the scanning tunneling microscopy and non-contact atomic force microscopy results in Fig. 5.

- 3. The paragraph discussing the Van der Waals interaction on page 12 is confusing to me. What is it precisely the authors intend to say. Likewise the message in the statement at the end of the paragraph is entirely obscure to me.**

In the original manuscript we wanted to compare the adhesion energy between graphite and graphene on SiO₂. However, we agree with the reviewer that this comparison was more of a general note *and we removed this paragraph from the revised version.*

- 4. The term Ne atom diffusion is misleading. The authors clearly say that they had to use imaging conditions corresponding to a short tunneling distance for observing this effect. I would rather use the term tip-induced translation in this context.**

We have addressed this issue. In the abstract, we changed “*the subsurface atom diffusion*” to “*the tip-induced subsurface translation of neon atoms*”, although we retained the term “*diffusion*” in Fig. 4.

Part II – Reviewer 2

Overall, the authors have performed careful experiments and analysis. The results are interesting and useful. The paper is clearly written.

- 1. However, my major concern is on the novelty and significance of the outcome of this work. It seems to be the measured adhesion energy of graphene that has been recently reported in a Science paper for mesoscopic graphite contacts (ref. [13]). Since the adhesion energy is a fundamental material property, it is expected to remain unchanged at the nanoscale, as verified by this work.**

Our goal here was to achieve nanoscale resolution in the measurement of adhesion energy for the case of van-der-Waals solids. We used graphite as a well-known material and showed that the single or few-atom intercalation method yields remarkably good quantitative agreement with macroscopic methods. It is relatively straightforward to envision extending this methodology to other 2D materials or to few-layer van-der-Waals heterostructures whose interlayer interactions have a dramatic effect on material and eventually device properties. Moreover, variation of the adhesion energy as a function of proximity to steps or defects can be also investigated with our methodology.

Even more generally, we believe that atom intercalation enables nanoscale strain engineering of layered solids, where the intercalate-mediated blisters act as so-called “Gaussian impurities”. Several theoretical papers [*Phys. Rev. B* **2014**, 90, 041411(R); *Phys. Rev. B* **2015**, 91, 161407(R)] proposed interesting electronic effects for such impurities in graphene, and in our work we show how such impurities can be introduced and studied.

We added a sentence about Gaussian impurities, with appropriate references, in the conclusion section.

2. In addition, it is not clear whether the implanted neon atoms will penetrate into several graphite layers. Is it possible that the multi-layered intercalates collectively produce the surface clusters?

This question is similar to the first one raised by the 1st referee. The detailed answer for this question is documented above (see our answer for question # 1 of Reviewer # 1).

Briefly, we carried out extensive DFT modeling of deep layer intercalation and additional experiments using, non-contact AFM technique for direct measurement of surface topography. Our overall conclusion is that deeper intercalation is in principle possible. It is, however, quite easy to identify sub-surface intercalates based on the height and curvature of blisters they create. Most importantly, the direct estimation of the adhesion energy based on the topographic measurement of the surface should be applied only to sub-surface intercalates (as we have in the original paper). Altogether this additional effort has significantly improved our analysis, and we are thankful to reviewers for their constructive criticism.

The following changes were made to the manuscript:

- (a) A new paragraph discussing the properties of deeper intercalates has been added on page 12.*
- (b) A new Fig. 5 comparing non-contact atomic force microscopy and scanning tunneling microscopy has been added.*
- (c) Detailed computational analysis of deeper intercalates (Fig. S3 through Fig. S6), and a Supplementary Note 2 have been added in Supplementary Information section.*
- (d) A new STM image of possibly deeper intercalates has been included as Fig. S1 in Supplementary Information section.*

3. In the abstract, what is the meaning for "measurement of elastic adhesion"? The "elastic adhesion" is not a commonly used term. In other words, what is "inelastic adhesion"?

The reviewer is correct. We have deleted the word “elastic” in the abstract.

Part III – Reviewer 3

As the author pointed out, the nano-scale measurement of adhesion is of great importance in understanding two dimensional materials. Previously blister tests were adopted to measure the graphene adhesion; however the measurement was confined to the μm -scale due to the spatial resolution of the atomic-force-microscopy. In this manuscript, the STM based measurement successfully observed the nano-size blister by the Ne atom intercalations. The proposed method was used to correlate the atomistic

simulation, and the measured data well agree with the reported values. Furthermore, it is quite interesting that the continuum mechanics model is still valid in atomic-size levels. However, some important details are lacking, and the following questions are raised.

1. What is the reason for choosing Ne? Would you be able to get the same adhesion energy with Ar intercalation?

We chose an inert gas so that the intercalated atoms do not form covalent bonds with HOPG. Ne was chosen over Ar, because it is a lighter atom and will be much less destructive upon ion sputtering. Nevertheless, even Ne is producing missing carbon defects. To clarify this point of the reviewer we added a new sentence in the Methods section (on page 15): “*Lighter neon was chosen over argon to minimize creation of point defects by sputtering.*”

2. As mentioned in the manuscript, Ne+ "ions" were sputtered to create the blisters and associated defects on HOPG. The ions are not inert and may strongly interact with electrons in graphene, greatly affecting the adhesion energy. However, in this manuscript, "inert Ne atom" was assumed not only for the analysis of experimental data, but also for the atomistic simulations. Please comment on this.

Ne⁺ ion will be promptly neutralized upon contact and intercalation with graphite. Neutralization of low energy Ne⁺ is well-known for metal surfaces with work functions in the range from 3-5.5 eV. [*Nuclear Instruments and Methods in Physics Research B* **2006**, 248, 16–20; *Surface Science* **2000**, 466, 127–136]. Moreover, our DFT calculations show minimal charge transfer between intercalated Ne atom and surrounding graphite lattice (<0.05e based on Bader charge analysis).

We reworded the text to correctly refer to Ne intercalates as atoms, and added the following sentences on page 5: “*Since we are working with a grounded sample, and both graphite and graphene are electronically conducting materials, Ne impurity is charge neutral. Neutralization of the low energy Ne⁺ is well-known for metal surfaces with work functions in the range from 3-5.5 eV.^{21,22} Moreover, our DFT calculations further confirm that there is a minimal (<0.05 e) charge transfer between intercalated Ne atoms and the surrounding C atoms of the graphite.*”

3. It was not clearly explained how the Ne+ ions were intercalated beneath the top graphene layer. Some questions related to this are as follows. Isn't it also possible that the Ne+ ions penetrate deeper and end up being intercalated beneath 2nd, 3rd or other layers?

A similar question was raised by both reviewers # 1 and #2. The detailed answer to this question of the reviewer has been presented above (See question #1 of Reviewer 1).

Briefly, we carried out extensive DFT modeling of deep layer intercalation and additional experiments using, non-contact AFM technique for direct measurement of surface topography. Our overall conclusion is that deeper intercalation is in principle possible. It is, however, quite easy to identify sub-surface

intercalates based on the height and curvature of blisters they create. Most importantly, the direct estimation of the adhesion energy based on the topographic measurement of the surface should be applied only to sub-surface intercalates (as we have in the original paper). Altogether this additional effort has significantly improved our analysis, and we are thankful to reviewers for their constructive criticism.

The following changes were made to the manuscript:

- (a) *A new paragraph discussing the properties of deeper intercalates has been added on page 12.*
- (b) *A new Fig. 5 comparing non-contact atomic force microscopy and scanning tunneling microscopy has been added.*
- (c) *Detailed computational analysis of deeper intercalates (Fig. S3 through Fig. S6), and a Supplementary Note 2 have been added in Supplementary Information section.*
- (d) *A new STM image of possibly deeper intercalates has been included as Fig. S1 in Supplementary Information section.*

- 4. Where did carbon atoms from the point defects go after being bombarded by Ne⁺ ions? Any possibility that the intercalated atoms are actually the C atoms instead of Ne atoms? The calculated energy barrier for the Ne atom diffusion is 0.14 eV; how about the energy barrier for the C atom diffusion in the gap of the graphene bilayer? Which atom is more likely to be intercalated in the gap?**

The carbon atoms are ejected into the UHV chamber. If C atoms were intercalated, they would form covalent bonds with graphene sheets, producing dramatic local changes topographically and electronically. Therefore we can reliably rule out their presence in the blisters which we used to quantify the adhesion energy.

- 5. The adhesion energy was calculated from the STM image at temperature $T = 77$ K, which is very low temperature considering real life applications of graphene. The manuscript says that the graphite adhesion energy of 0.221 ± 0.011 J/m² determined in this study is in excellent agreement with reported macroscopic values and the atomistic simulations of this study. This reviewer wonders that whether the values being compared were also measured or calculated at the same temperature $T = 77$ K. Please add some comment on the temperature dependence of the adhesion energy of graphene. Is the adhesion energy at room temperature higher or lower than that at 77 K?**

Our measurements at 77K in a controlled environment at atomic scale enable direct verification by ground state DFT calculations. In addition, our 4K and 77K measurements yield consistent parameters for the blister. At the same time, in a recent paper [*Nano Lett.*, **2016**, 16 (1), 387–391, by using molecular dynamics to investigate the temperature dependent adhesion of a suspended graphene sheet on a trench of

15 nm in width] it was found only sub angstrom changes of the depth of the graphene sheet in the temperature range 0-2000K. We therefore believe that adhesion energy will be closely comparable between 77K and 300K.

To clarify the question of the reviewer we added a new sentence in the Methods section: “*The STM profile measurements over different blisters taken at 77 K and 4.3 K yield consistent results, suggesting that in this temperature range the adhesion energy shows little change, as may be expected (see Supplementary Note 3).*”

- 6. Can the intercalation and STM measurement be applied on other substrates? For example, can the adhesion energy of transferred graphene on SiO₂ be measured by the method suggested in this study?**

We envision that this methodology can be generally applied to other substrates. Also, we are currently setting up similar experiments for other 2D materials.

- 7. Term and symbol should be appropriately used. In case of the E2D, it is the "two dimensional" Young's modulus (the unit is different with the Young's modulus). In case of the adhesion energy, gamma (Γ) is suitable rather than E.**

We have added “two dimensional” in front of the “Young’s modulus” on page 6 where E2D term is defined. We have replaced E with the gamma (Γ) notation for the adhesion energy in the text.

- 8. How was the blister radius (a) defined in case of the triangular blister? (Fig. 3a)**

The “triangular” shape term mentioned here was used for easy identification. To determine the radius of such blisters, we approximated them as having Gaussian curvatures, similar to the other blisters. We have added a brief discussion of this point in the Supplementary Note 1.

REVIEWERS' COMMENTS:

Reviewer #1

In comments made directly to the editor Reviewer #1 stated that they were satisfied with the revisions and recommended that the manuscript be accepted.

Reviewer #2 (Remarks to the Author):

The authors have clearly made significant efforts to address the review comments from three reviewers. My questions have been appropriately answered. I have no further comments that preclude publication of the paper in its current form.

Reviewer #3 (Remarks to the Author):

The authors responded the reviewer's comments and revised the manuscript satisfactorily. I recommend its publication in Nature Communications.